



# The 1921 European drought: Impacts, reconstruction and drivers

Gerard van der Schrier[1], Richard P. Allan[2], Albert Ossó[3], Pedro M. Sousa[10], Hans Van de Vyver[4], Bert Van Schaeybroeck[4], Roberto Coscarelli[5], Angela A. Pasqua[5], Olga Petrucci[5], Mary Curley[6], Mirosław Mietus[7], Janusz Filipiak[8], Petr Štěpánek[9], Pavel Zahradníček[9], Rudolf Brázdil[9,10], Ladislava Řezníčková[9,10], Else J.M. van den Besselaar[1], Ricardo Trigo[11], and Enric Aguilar[12]

[1]Royal Netherlands Meteorological Institute, De Bilt, the Netherlands
[2]University of Reading, United Kingdom
[3]Wegener Center for Climate and Global Change, University of Graz, Austria
[4]Royal Meteorological Institute, Uccle, Belgium
[5]National Research Council of Itly, Research Institute for Geo-Hydrological Protection CNR-IRPI, Rende, Italy
[6]Met Éireann, Dublin, Ireland
[7]Institute of Meteorology and Water Management, Warsaw, Poland
[8]Department of Meteorology and Climatology, Institute of Geography, Faculty of Oceanography and Geography, University of Gdańsk, Gdańsk, Poland
[9]Global Change Research Institute, Brno, Czech Republic
[10]Masaryk University, Brno, Czech Republic
[11]Instituto Dom Luiz (IDL), Faculdade de Ciências, Universidade de Lisboa, 1749-016 Lisboa, Portugal
[12]Center for Climate Change (C3), Universtitat Rovira i Virgili, Tarragona, Spain

**Correspondence:** G. van der Schrier (schrier@knmi.nl)

**Abstract.** The European drought of 1921 is assessed in terms of its impacts on society and in terms of its physical characteristics. The development of impacts of the drought are categorized by a systematic survey of newspaper reports from five European newspapers covering the area from England to the Czech Republic and other parts of Europe. This is coupled to a reconstruction of daily temperature and precipitation based on (rescued) meteorological measurements to quantify the drought

5    severity and extent, and reanalysis data is used to identify its drivers. This analysis shows that the first impacts of the drought started to appear in early spring and lingered on until well into autumn and winter, affecting water supply and agriculture and livestock farming. The dominant impact in western Europe is on agriculture and livestock farming while in central Europe the effects of wildfires were reported on most often. The peak in the number of reports is in late summer. Preceeding the first impacts was the dry autumn of 1920 and winter 1920/1921. The area hardest hit by the drought in the following spring and

10    summer was the triangle between Brussels, Paris and Lyon, but a vast stretch of the continent, from Ireland to the Ukraine, was affected. The reported impacts on water supply and water borne transport in that region were matched by an analysis of the hydrological situation over the Seine catchment. On average, the 1921 summer was not particularly hot but the heat wave which was observed at the end of July saw temperatures matching those of the heatwaves in modern summers. Similar to modern droughts, an anticyclone was present roughly over the British Isles, maintaining sunny and dry weather in Europe and

15    steering away cyclones to the north. Its persistence makes it exceptional in comparison to modern droughts.



The 1921 drought stands-out as the most severe and most wide-spread drought in Europe since the start of the 20th century. While none of the seasons in 1920 and 1921 tops the scale of having the largest precipitation deficit on record, the conservative nature of drought amplifies the lack of precipitation in autumn and winter into the following spring and summer.

## 1   Introduction

Drought is one of the most costly natural disasters (Van Lanen et al., 2016). The severe drought that affected large areas of Europe in 2018 resulted in widespread losses in agriculture and forestry and produced an overall loss of around €3.3bn, making it the year's costliest event in Europe (MunichRe, 2019). A comprehensive view of past drought events based on the long instrumental records extending back into the 19th century is, however, far from complete yet highly essential in order to put the recent European droughts (2003, 2010, 2013, 2015, 2018) and future ones into perspective (Hanel et al., 2018). Changes in droughts in a future warming world may vary strongly between regions and models (e.g. Cook et al. (2019); Naumann et al. (2018); Stagge et al. (2017); Trenberth et al. (2014)), but, models consistently project more frequent and severe droughts over the Mediterranean region (Spinoni et al., 2018). A deep understanding of past extreme events matters from an attribution perspective and recently different frameworks have been proposed to address the anthropogenic influences on specific events (e.g. Zahradníček et al. (2015); Brázdil et al. (2016, 2019); Harrington et al. (2019)).

Droughts are often viewed from a climatic perspective where a host of indicators are available based on meteorological parameters (e.g. McKee et al. (1993); van der Schrier et al. (2006); Vicente-Serrano et al. (2010)). A more systematic way of combining the physical and impact dimensions of drought is necessary (Stahl et al., 2016), particularly for historical events from the first half of the 20[th] century which are never assessed in detail, often due to the lack of sufficient meteorological and documentary data. The aim of this study is therefore to provide a more holistic description of the exceptional drought that hit Europe in 1921. An overview of the impacts which this drought had on the various sectors of European society is presented, along with a reconstruction of the severity and spatiotemporal drought extent in terms of meteorological and climatological indices and a description of the drivers of this drought.

The principal motivation to focus on 1921 is that this 100-yr old climatic extreme is the driest year since the start of the 20[th] century for Europe in terms of area affected (van der Schrier et al., 2006; Hanel et al., 2018). The drought of 1921 affected a large part of Europe and extended east up to the Volga region and Ural mountains where it contributed to the famine in Russia of 1921-1922 (Fisher, 1927). With the recent dry years in Europe (Rebetez et al., 2006; Barriopedro et al., 2011; Buitink et al., 2020) and the discussion in the literature on a possible increase in drought frequency or severity (Trenberth et al., 2014; Spinoni et al., 2018), a complete and detailed record of historic droughts is needed to put recent events in perspective. An early overview of the 1921 drought for Europe is provided by Eredia (1923) of which an English summary is provided by Bonacina





(1923), for the British Isles by Brooks and Glasspoole (1922) and for the French Dauphine region by Blanchard (1922). More recently Brázdil et al. (2015) analysed the 1921 drought for the Czech Republic.

## 2 Impacts of the 1921 drought

This section describes the 1921 drought in terms of its consequences on socioeconomic and environmental systems, i.e. its

negative impacts (Blauhut et al., 2015). The information used in this section is almost exclusively based on text-based reports. The 1921 drought is described by its physical characteristics, like the reduction of precipitation, the loss of moisture to the overlying atmosphere and the associated increase in air temperature, in section 3 and~4.

### 2.1 Impacts analysis on text-based reports

In a systematic effort to review the impacts of the 1921 drought, text-based reports were gathered and the reported drought

impacts were classified into major impact categories, each of which had a number of subtypes. The distribution of these categories and types was then analyzed over time for five countries in Europe. Different selected newspapers were explored through their digital archives by means of the search term 'drought' (in each of the native languages). The resulting newspaper clippings were then categorized, following Stahl et al. (2016, their Table A2) and labelled by area or place and date (on a weekly time scale). Text-based reports from the following digitized newspapers were used: the Birmingham Gazette (United

Kingdom), the Algemeen Handelsblad (the Netherlands), the Standaard (Belgium), the Berliner Tageszeitung (Germany) and the Lidové noviny (the Czech Republic).

Figure 1 shows the impact-categorization statistics for the five countries aggregated in 7-day periods (Sunday-Saturday). The colour coding relates to the main impact categories defined by Stahl et al. (2016). Table 1 summarizes, for each of these countries, the earliest, latest and top-3 most reported impacts. The most frequently covered impacts across countries relate to

agriculture and livestock farming with the exception of the Czech Republic where this impact ranks only 3rd and wildfires are reported most often. An indirect illustration of the impact of drought on agriculture is provided by photograph A5, showing childern playing with leaves that have been shed in the period of drought. Such scenes are also witnessed in city parks in the Netherlands [1] and in the Dutch apple orchards [2].

The most reported impacts that occur the earliest in the year, are different among the countries, but both BE and DE report

the earliest on effects on waterborne transportation. Photograph A1 documents the low water levels in the Rhine falls near Neuhausen (Switzerland) in March 1921 which will be illustrative for other rivers in the area. For most countries, the effects on water supply and agriculture and livestock farming persist until the end of the year.

The period with the largest number of reports is in July 1921. For the United Kingdom and Belgium this peak takes place in the week of 17 July, the Netherlands (NL) has the largest amount of reports in the week of 24 July and Germany (DE) and

the Czech Republic (CZ) in the week of July 31, showing a clear west-east time shift. The timing of these reports coincides

---

[1] de Telegraaf, 21-07-1921; Algemeen Handelsblad, 26-07-1921

[2] Algemeen Handelsblad, 10-07, 22-07 and 4-8-1921





**Table 1.** Earliest and latest reported impacts for the five countries, and the top-3 of most reported impacts for 1921.

|  | earliest | 1st | 2nd | 3rd | last |
|---|---|---|---|---|---|
| UK | agriculture and livestock farming | agriculture and livestock farming | wildfires | human health | water supply |
| NL | wildfires | agriculture and livestock farming | water supply | wildfires, human health, water quality | water supply |
| BE | waterborne transportation | agriculture and livestock farming | water supply | wildfires | human health, agriculture and livestock farming |
| DE | waterborne transportation | agriculture and livestock farming | wildfires, waterborne transportation | energy and industry, water supply | agriculture and livestock farming |
| CZ | soil system | wildfires | water supply | agriculture and livestock farming | water supply |

with the heat wave peak in Europe, with temperatures peaking in London around July 19 and in Paris (and in about 75% of France) on 28 July to 38.4°C, the latter not seen since 1881[3] (more on the heatwave in Sect. 4.4 and Fig. 10). Figure 1 also shows anomalous daily maximum temperature and rainfall ratios (with respect to the 1981-2010 climatology), averaged over the respective countries, superposed on the impact information. For all five counries, the peaks in reported impacts follows

periods of sustained high temperatures or peaks in temperatures. Similarly, the the peak in reported impacts are at the end of a long period of below normal precipitation, although the length of the dry spell is varies from country to country. The countries with a long period of below-normal precipitation show a more gradual build-up of the number of reported impacts (like GB, NL, BE) whereas DE en CZ show a more sudden increase in reports following a relatively short dry period. Less clear is the relation in a decrease in reported impacts with the decrease in temperatures and the shortlived wet period in August.

Impacts on waterborne transportation are reported on relatively often in Belgium and especially Germany and not so much in the Netherlands. Although the water level in the Rhine (where it enters the Netherlands) has been record low from spring to autumn 1921, it remained possible to navigate the Dutch part of the river (Rijkswaterstaat, personal communication). However, the Meuse, with its catchment in France, Belgium and the Netherlands was completely dry in the southeastern part of the Netherlands [4]. Photograph A4 shows that in the north of the Netherlnds, the canals were completely dry at the end of summer.

The impact categorization scheme (Stahl et al., 2016) gives the main impact category and a more detailed impact type within each category. This facilitates further differentiation of impacts. Figure 2 shows for the five countries within 'Agriculture and livestock farming' the eight impact types and for 'Public water supply' the seven impact types. This shows that the impact

[3]Le Figaro, 29-07-1921
[4]de Telegraaf, 15-07-1921



'Reduced productivity of annual crop cultivation' dominates the agricultural impacts. The decision to sell livestock in response to the drought was reported particularly often in the Czech Republic. For the 'Public water supply', limitations of use to
households (combined rural and in urban areas) was reported on most often. Effects of limited supply of drinking water are illustrated in photographs A2 and A3, where water from smaller rivers in central and southeast Netherlands is used as addition to the water supply in households.

In many reports, particularly those from the Netherlands and England, the late spring frost of 1921 was mentioned as an additional setback for farmers. However, an analysis of the Julian day with the last night with frost using the meteorological
station data, did not show a significant difference between the 1921 last spring frost and 1981-2010 climatological mean values of the last spring frost.

## 2.2 Recorded regional impact of the 1921 drought

### 2.2.1 North Italy and the Alpine region

The 1921 drought in Italy was preceded by dry conditions in 1920 and the first reports on its impact appeared in mid-November
1920 on a lower harvest for some products, especially cereals, due to the persistent drought. The dry conditions starting in autumn 1920 led to a dramatic decrease of water levels in Italian rivers, streams and lakes, especially those that came from the Alps. Recently, a reconstruction of monthly precipitation of the Abba basin in northern Italy for the period 1800-2016 (Crespi et al., 2021) confirms these reports as the annual precipitation sum over the Abba basin was observed to be the lowest on record. Dry conditions in the Alpine region are reflected by the situation in Switzerland where, following a rainy summer in
1920, a deficiency of precipitation started in winter 1920/21 and continued until the 1921 autumn. Salter (1921) wrote that the 1921 drought in Switzerland was the driest period since 1788.

Interestingly, the dry winter 1920/21 in Switzerland was reflected in an impact on the tourism sector; the Birmingham Gazette of March 16, 1921 reports on "dissappointed tourists [which] are leaving the country in large numbers".

In January 1921, hydro-powered electricity production was scarce because of the low water levels in dam reservoirs and
the problem intensified in autumn 1921 and winter 1921/22. Several provinces of northern Italy established severe reductions on the use of electrical energy, for industries, homes, cinemas, public/private offices and shops. This meant that major cities in northern Italy were without light or light was rationed and industries and transportation (locomotives) ceased operation. The water shortages at the start of 1921 affected the agricultural sector: cattle were sold before they died of thirst and lack of fodder. Drinking water became scarce, with queues for public fountains, and villages were waterless, requiring marches
of several kilometers to fetch water. A review in La Stampa of mid-December 1921 reports that throughout Upper Italy the water reserves are close to exhaustion impacting on hygiene and deteriorating the quality of the water. Large scale wildfires are observed, with complete hills prey to the fires. In the Trentino region, the substantial decrease of water level in a lake exposed a white stone put on a rock with an inscription in remembrance of the low lake level during the drought event of 1806. The water level in the "Lago Maggiore" lake decreased by about 2 m below normal level in this period, forcing the steamer lines to
change the landing place in several stations around the lake.





### 2.2.2 Ireland

At the end of April and early May, reports began to appear in the Irish newspapers that water reservoirs and rivers were lower than normal for the time of year and the public were urged to economise water usage as much as possible. By June the drought had started to affect livestock, grass growth was poor and crops in many parts of the country were reported to be suffering. In

the second half of June there continued to be anxiety over the development of crops due to the almost entire absence of rain, the late spring frosts and the low night temperatures. There were reports of water supplies being shut down in some parts of the country at night and by the end of the month there were serious concerns in Dublin that the Vartry reservoir would run dry. In July, newspaper reports stated that the drought still prevailed in all parts of the country and there were fears of water famine. Further appeals were made for citizens to be strictly economical in the use of water. Ponds and rivers that had not been seen

dry by the oldest inhabitants, were completely dried up in some areas. Thousands of cattle had to be driven for miles daily to be watered and many cows in all parts of the country were ceasing to give milk.

    The sunny and stable weather made for an early harvest. In some cornfields bordering the sea in the Hook, Co. Wexford district, the oat production was so poor owing to the great drought that the crop had to be cut and gathered up like hay. However, the drought did not affect the winter wheat crop contrasting with the situation of the pastures which was hardly short

of disastrous, particularly for grazers

### 2.2.3 France

The dry period which started in the last quarter of 1920 had the consequence of a general lowering of groundwater tables and a significant decrease in the flow of rivers. An example is the Seine basin where the small rivers in the Champagne region ran dry and in particular the Somme; the source of which has been dry since September 20, 1920 (P., 1921). The reduction

in streamflow impacted on the extensive canal system in northern France causing serious inconvenience in the operation of inland waterways. Traffic on the canal de Bourgogne stopped and traffic on the Rhône-Rhine canal between Strassbourg and Montreux-Chateau was strongly reduced (P., 1921).

    The succession of rainfall deficits for 5 consecutive quarters in France, with the autumn of 1921 being particularly dry, had repercussions on the production of hydroelectric energy and on the supply of drinking water for large cities like Grenoble,

Chambéry, Gap due to rapid and significant drops in underground aquifers deprived of recharge (Duband et al., 2004). Duband et al. (2004) observed that 1921 had the lowest annual flow for the Rhine and Rhone and 2nd lowest for the Loire and Seine (1949 was slightly lower) in a 150 year record.

    The dry conditions in summer 1921 coincided with a heatwave, which will be quantified in Sect. 4.4. The newspaper Le Figaro (July 29, 1921) describes a short-lived wind from the south advecting hot and dry air into the capital, in the height of

the heat wave. To protect themselves from the heat, Parisians held wet clothes before their faces and wore wet shirts.





### 2.2.4 United Kingdom

Publications following the 1921 drought highlight the scale and severity of this event. The rainfall deficiencies for 1921 in eastern and southern England amounted to between 50% and 60% of the normal value (Office, 1922). The Met Office report for the rainfall for 1921 argues that it was the most remarkable for the period of about 100 years over which rainfall has been recorded, and that 1921 was the driest year in London for practically a century and a half. Over the Thames Estuary rain fell on less than 100 days during the year. The impact of the exceptionally low number of rainy days is documented in photographs A6 and A7 showing low water levels in the Thames estuary during the second half of August 1921.

The overview of London weather from 1841 to 1949 (Marshall, 1952) had 1921 as the driest year by far with only 100 mm rainfall. Only January had normal rainfall amounts. Partial drought conditions (in which mean daily rainfall does not exceed 0.4mm) span the period from May 10 until August 12. The drought peaked between June 4-19 and September 20-October 13 when precipitation on each of these days was below 0.4mm. These contemporary reports are confirmed by a review of major droughts in England and Wales, in which Cole and Marsh (2006) classify the 1921-1922 drought as a 'Major Drought' with the second lowest 6-month and third lowest 12-month rainfall totals for England and Wales and affecting most in the UK. The drought had major soil water impacts in the latter half of 1921, heralding low ground water levels in 1922.

The film news of Pathé (British Pathé, 1921) vivdly presented the consequences of the drought of 1921. The film is produced on July 28th, 1921, a week after newspapers reported an 'unparalleled heat wave' on July 19[5] which had temperatures in excess of 30°Cin the West Midlands, the South west, South East and the East of England. The film bears the title 'The peril of the drought' and the subtitle ('Burning crops and stacks menace our winter food supplies') reflects the fears that the British were at risk of being deprived of their harvests for the coming winter, with the food rationing still fresh in their memories. The movie shows fields and haystacks on fire as firemen try to put them out. Wildfires are also reported on most often in the British written press (Fig. 1) and photograph A8 documents a scene with several wildfires in Northern Surrey during early September.

### 2.2.5 Poland

A striking impact of the 1921 drought in Poland were its many fires, the of which occurred in August. These include the one on August 7 in the town of Pińsk (population in 1921: 23.497), eastern Poland, where loss of life was reported and a third of the town, predominantly built of wood, was destroyed, including the town centre and an almost 1000-year-old synagogue. The losses were so great that the Polish Government decided to move the authorities of the region to which Pińsk belongs to Brest, located 150 km westward. On the same day, the smaller town of Kłodawa (population in 1921: about 2.200) in central Poland, saw a large part of the town completely destroyed by fire where at least 130 buildings burned down. The town of Rudnik on the San (population in 1921: 2.959) in the southern part of Poland was densely covered in smoke from the fires in the surrounding bogs, meadows and forest, making it difficult to breathe.

Between 9-16 August 1921, forest fires raged in the Sandomierz Forest in the southern part of Poland. At least a dozen square kilometers burned down in this fire. For August 13, reports on forest fires in the region of Upper Silesia are found. The

---

[5]The Register (Adelaide) 20/07/1921





fires have been estimated as the biggest ones in the region of Silesia for at least several dozen years. Not only many square kilometers of forests but also suburban districts of the town of Mikołów (formerly: Nikolai) were burned down. The railway

traffic on the route Zabrze/Gliwice - Kędzierzyn (formerly: Hindenburg/Gliwitz - Kendrin) was temporarily suspended due to the risk of flames.

An assessment of the production of annual crops in Western Poland on August 1, 1921 showed that many crops for human consumption or fodder rated between 'medium' and 'poor' up to 'bad' for potatoes, sugar beet and pastures. Widespread deficits in the availability of dairy products due to a strong reduction in milk production was reported, as well as low availability of

meat and sausages, which resulted in price increases, particularly in Western and Central Poland. Restrictions in drinking water were issued in Poland as well.

## 3  Description and Quantitative analysis of the 1921 drought

To further develop and compliment the qualitative assessment of impacts related to the 1921 European drought, we now provide quantitative assessment of this climatic event using new datasets and methods.

### 3.1  Data gathering and data rescue

An initial assessment of the number of stations in the European Climate Assessment and Dataset (ECA&D) (Klein Tank et al., 2002) for 1921 showed that the coverage was poor to the point that a reliable reconstruction of pan-European precipitation was not really possible. The quality of a gridded dataset is largely dependent on the amount of data that is used to construct the data (Cornes et al., 2018), which motivated us to increase the station coverage for this historic period. The reasons for using a

gridded dataset rather than the original stations are the possibility to better assess the spatial scale of the extreme event more easily and its ease of use for the wider research community.

Three approaches are used to increase the station coverage for 1920-1921 in ECA&D.

1) Within the ERA4CS INDECIS project[6], a data-rescue effort was made for data from the Balkans and Italy (two areas which have a particularly low station density in ECA&D) and the Irish Meteorological Service Met Éireann digitized all data

for four long-running stations. The Polish Institute of Meteorology and Water Management (IMGW) digitized data from the current-day Polish borders and within former Polish regions which now belong to Belarus and Ukraine, thereby adding 18 temperature-recording stations and 79 rain-gauges.

2) The data portals of the European National Meteorological Services (NMSs) were accessed in search of data for 1920 and 1921 that was not already included in the ECA&D database. This approach was directed mainly at the Scandinavian countries,

Germany and the UK.

3) Inquiries were made at the NMS contacts of ECA&D with a request for additional data. The Czech Hydrometeorological Institute (CHMI) provided data from 8 stations which continued from 1920 to 1960. In addition, a set of 359 stations with data

---

[6]www.indecis.eu





from 1920 and 1921 has been provided. A set of 187 rain gauge stations were provided from the Dutch NMS. These latter data provisions are for stations that are now not longer operational.

Figure 3 shows the station coverage that is used for the current study. For some countries, such as Poland, much data still remains to be digitized (and is not digitally available to both the NMS and ECA&D).

## 3.2 Construction of the gridded E-OBS for 1920-1921

The aforementioned station dataset is used to develop the E-OBS dataset for 1920 and 1921. The gridding procedure was developed by Cornes et al. (2018) and adopts a two-stage process to produce the daily fields. The spatial trend in the daily temperature variables is captured by fitting a Generalized Additive Model (GAM) to the station values, where the daily temperature data are modelled as a smoothed function of longitude, latitude, and altitude using a reduced-rank thin-plate spline, plus a smoothed function of the monthly mean, background field values of temperature using a cubic spline. For precipitation, altitude is not used in the spatial trend as this did not significantly improve the fitting of the model (Cornes et al., 2018). Note that the rain gauges in areas with complex topography in the 1921 situation are likely to be concentrated in the valleys. To remove some of the skewness in the data, the precipitation values and monthly background precipitation totals were square root transformed prior to fitting. Step two is to interpolate the residuals from this model using a stochastic technique (Gaussian Random Field simulation) to produce the daily ensemble. The occurrence of precipitation (as a binary field) was gridded separately from accumulations using a full thin-plate spline. This was then used to mask the daily fields.

Figure 4 shows the coverage of the grids for precipitation and temperature for 1920 and 1921 and the number of days for which data is available. For temperature the eastern part of the Mediterranean is missing as well as an elongated stretch from Belarus into Russia. For precipitation, the largest part of Europe is almost fully covered except for a few isolated spots in more remote areas that do not have full time coverage. Iceland is the exception: the data for 1920 is much less complete than for 1921. A stretch of data from Belarus into Russia is missing for 1920, but for 1921 this is already much better. The E-OBS gridded fields for 1920 and 1921 comes in the grid resolutions of 0.25° and 0.1°, covering the area 25°N-71.5°N× 25°W-45°E.

## 3.3 Other data used for reconstruction

In the analysis, a long-term daily precipitation series from ECA&D are used (Klein Tank et al., 2002) for Uccle (Belgium), Marseille (France), Milan (Italy), and St. Petersburg (Russia). The E-OBS reconstruction for 1920-1921 is combined with E-OBSv20.0e covering the 1950-2018 period for e.g. the calibration of the climate indices. For the synoptic analysis we select monthly-mean data from the ERA20C reanalysis product (Poli et al., 2016) which covers the 1900-2010 period and has a horizontal resolution of about 125 km. The following variables were retrieved for pressure levels between 1000hPa and 300hPa: geopotential height (Z), zonal (U) and meridional (V) wind and specific humidity (q). Geopotential height at the 500hPa level will be referred to as Z500 hereafter. Sea Surface Temperature data (SST) was retrieved from the Met Office Hadley Centre HadSST dataset (Kennedy et al., 2019).





## 3.4 Methods

There are numerous ways to quantify drought (e.g. Heim (2002)) where indices are based on precipitation alone or can include the storage of moisture in the soils, or the loss of moisture to run off or the overlying atmosphere. Here we opt for using a straightforward index that relates to the precipitation amount and standardizes the metric against local climatic conditions, and use other means to quantify the loss of moisture to the atmosphere. The transport of moisture to (or away from) the European continent relates the drought to atmospheric circulation and a heatwave index is used to quantify the heat during the 1921

summer.

### 3.4.1 Standardized Precipitation Index

A common meteorological drought index to analyse drought conditions is the standardized precipitation index (SPI McKee et al. (1993)), which quantifies the anomaly of an observed accumulated precipitation depth over the time scale of interest (e.g. 3, 6, 12 months), calculated using a long-term time series. The cases SPI > 0, SPI < 0, and SPI = 0 indicate wet, dry, and

normal conditions with respect to the reference climate, respectively. Another useful precipitation index to quantify drought is the number of rainy days with totals $\geq$ 1 mm (RR1).

### 3.4.2 Estimates of evaporation

The evaporation of water from the soils is principally driven by radiation which provides the energy for evaporation. As the observation-based reconstruction of the 1920-1921 climate lacks an estimate of daily global radiation sums, a commonly-used

alternative approach uses the Hargreaves equation for potential evaporation $ET_0$ (Hargreaves and Samani, 1985; Hargreaves and Allen, 2003). The relation between sunshine data with the diurnal temperature range provided the possibility to construct an estimate for evaporation that is based on maximum and minimum temperature (which are available in this reconstruction).

The form of the Hargreaves and Samani (1985) parameterization used here is:

$$ET_0 = 0.0023 Ra(Tc + 17.8)\sqrt{DTR} \tag{1}$$

where $Ra$ is the extra terrestrial radiation (which is a function of latitude and yearday in $W/m^2$), $Tc$ is the average daily temperature and $DTR$ the diurnal temperature range, both in $°C$. The empirical coefficients were determined in various field experiments (Hargreaves and Allen, 2003). By comparing daily $ET_0$ values from 1921 as a ratio of present-day values we aim to mitigate some of the unavoidable deviation from more realistic estimates of daily $ET_0$. Here the climatological period 1981-2010 is used as a reference period.

### 3.4.3 Integrated Vapor Transport

Moisture availability and transport were evaluated by using the vertically integrated water vapor transport (IVT), based on ERA20C data, which is defined as the horizontal transport of specific humidity, integrated for a vertical column of the tropo-





sphere (between 1000 and 300 hPa), as follows:

$$IVT = \sqrt{\left(\frac{1}{g}\int\limits_{1000hPa}^{300hPa} qu\mathrm{d}p\right)^2 + \left(\frac{1}{g}\int\limits_{1000hPa}^{300hPa} qv\mathrm{d}p\right)^2} \tag{2}$$

where $q$ is the specific humidity $[kg/kg]$, $u$ and $v$ the zonal and meridional layer averaged wind $[m/s]$, d$p$ is the pressure
difference between two adjacent levels $[Pa]$ and $g$ is the acceleration due to gravity $[m/s^2]$.

### 3.4.4   The metric for warm day-times and the length of heatwaves

The heatwave index used here is the Warm Spell Duration Index (WSDI) defined by the CCl/CLIVAR/JCOMM Expert Team on
Climate Change Detection and Indices (ETCDDI) (Klein Tank et al., 2009). The WSDI relates to the locally defined threshold
of the 90[th] percentile in daily maximum temperature. Following the definition of this index, at least six days in a row need to
exceed the 90[th] percentile threshold for a warm spell to be observed, where the 90[th] percentile is based on data, smoothed using
a 5-day running mean, from the 1961-1990 period. A day where the daily maximum temperature exceeds the threshold of the
90[th] percentile is called a 'warm day-time' (as opposed to a warm day which relats to the daily-mean temperature and a warm
night which relates to the daily minimum temperature). The WSDI is then simply a string of consecutive warm day-times.

## 4   Results

### 4.1   The 1921 drought in the long-term context

The SPI indicator computed for the accumulation period of 12 months is shown in Fig. 5 for Uccle, Belgium, based on the
precipitation series starting in 1880. The drought in 1921, with a value below -4, clearly stands out in comparison to more
recent droughts, indicating the exceptional nature of this period. For comparison: peak values in the dry year 2018 are -2.65 for
SPI3 (July 2018, accumulating precipitation for May-June-July) and -1.87 for SPI6 (Oct. 2018, accumulating precipitation for
May to Oct.) and peak values in 1976 are -3.44 (June 1976) and -2.84 (June 1976). For the calculation of the SPI values, the
calibration interval 1950-2019 is used. The overview of drought severity of major European cities in Fig. 6 shows that Uccle is
in the area in Europe where the 1921 drought was the longest and most severe, together with Paris and Lyon.

### 4.2   Spatial variations in the 1921 drought for Europe

Figure 7 shows the spatial distribution of the SPI-values over Europe as the drought evolves from autumn 1920 to autumn 1921,
based on the E-OBS dataset. The SPI3 values for Sep-Nov 1920 (Fig. 7a) show dry to extremely dry conditions in a large area
from England and Wales eastward, including southern Scandinavia, northeastern Europe and the northern part of the Balkans.
The driest spots are found in the Netherlands and northern Germany, southern Sweden, Estonia and northwestern Russia.
Mediterranean Europe is rather wet, but with moderately dry conditions in northeast Italy. This figure shows that the end of
1920 was already particularly dry. The dry conditions are less expansive in the 1920/1921 winter (Fig. 7b), while particularly



France, Ireland, the Ukraine and some more isolated areas in Europe remained in a droughty state. In spring 1921, the drought shifted a bit to the south-west with the focal point over Brittany in western France, but the SPI values remained well below -1 or -1.5 for a large part of Europe (Fig. 7c). The SPI3 map for summer 1921 (Fig. 7d) shows a further intensifying drought, especially in southern England and Belgium. The wet conditions in Mediterranean Europe have moved away, except in south
and central Italy and along the Adriatic coast in the Balkans. In autumn 1921 (Fig. 7e), the drought further intensified with the driest points in central France and northern Italy and south Sweden. A large part of west and central Europe remained firmly in its grip.

The assessment of the drought conditions on a monthly basis (not shown) shows that the driest months of 1920 were October and November, whereas the driest months of 1921 were March and July. There were, however, a lot of subsequent dry months
in 1921, and only August showed somewhat normal conditions. The remarkable nature of 1921 is made clear in Figure 8 showing, for each location in Europe, the probability of the number of rainy days in 1921, where the 1981-2010 period is the baseline period. This figure shows large areas of western and central Europe, extending toward the east, with a very low probability indicating that the number of rainy days in 1921 was well outside the distribution in the baseline period.

### 4.3 Rainfall and evaporation over river catchments

In the inventory of impacts of the 1921 drought, the limitations in the water supply for drinking and sanitation purposes and to a lesser extent the limitations in waterborne transport ranked high. An assessment of the remarkable nature of the 1921 drought can be made by making a cumulative distribution of the daily rainfall, integrated over a river catchment. In an effort to assess the loss of evaporation to the atmosphere, a similar integral measure over a river catchment of potential evaporation is made. Figure 9 shows this analysis for the Seine catchment. The left panel show the cumulative distribution of daily rainfall
for the combined years 1920 and 1921 (in red) and the individual years in the 1981-2010 period (in grey). The red line is outside the ensemble of climatological years, indicating that the number of dry days was higher than what was seen in the baseline period 1981-2010 and that there were much more days with low amounts of precipitation in 1920 and 1921 than in the modern climatology. The lack of rain was aggravated by anomalously high amounts of potential evaporation. The right panel of Figure 9 shows daily values of the 1921 $ET_0$ as the ratio of the climatological 1981-2010 value. The aggregated potential
evaporation over the Seine is for 81.4% of the time larger one, exceeding the mean 1981-2010 value, and for 47.1% of the time above the 90th percentile, whereas it drops below the 10th percentile for just 2.5%. The highest peaks in potential evaporation (with respect to the climatology) are observed in July and in the autumn where values peaked to over twice the climatological value. Similar observations apply to the Moselle catchment (not shown).

### 4.4 Heatwave quantification

The 1921 drought coincided with high temperatures across Europe and parts of which were affected by a heat wave which occurred in July. While the European-average July temperature was not remarkable in the 100-year period 1920-2019 (mean daily average temperature ranks 47th) , the hottest day in summer 1921 was remarkable. When compared to all other hottest summer days in the period 1920-2019, 1921 ranks in the top 5 for the area between Paris, eastern half of Belgium, southwest



Germany and into Switzerland. The hottest day of 1921 also ranks in the top 5 of the past 100 years in western Poland, the
Black Sea coast of Romania, western parts of the Ukraine, parts of Ireland and the North Sea coast of northern England.

Figure 10a shows the spatial pattern of the Warm Spell Duration Index and gives the length of the longest warm spell
observed in summer 1921. Figure 10b shows the number of warm day-times in summer 1921 (regardless if these days are
clustered or not). Figure 10 shows that the 1921 heat wave covered several regions of a large latitudinal band stretching from
Ireland to Ukraine, including southwest England and Wales, central France and a large area in central Europe and the Balkans,
where a long period of persistently high temperatures can be observed. Finally, a large sector of Ukraine and western Russia,
located just north of the Black Sea observed a relatively short duration warm spell. This does not mean that high temperatures
were not observed in other parts of Europe. Figure 10b shows that also outside the heat wave-stricken areas a large number of
days in summer 1921 exceeded the 90th percentile - more than the expected 9 days if this was an ordinary summer.

One of the hottest days in the heatwave that affected France was July 28, 1921. Figure 11 shows the daily maximum
temperature for this day, where temperatures in France locally exceeded 40°C, and about three quarters of France, including
Paris, saw temperatures of 38°C or higher (briefly discussed in Sect. 2.2.3. These temperatures are high for today's standards
as well. Ranking the highest daily maximum temperature reached in summer 1921 in the 100-year E-OBS record shows that
a sizeable part of France, from Paris into Switzerland, have the highest 1921 summer temperature in the top-5. Further east in
an arc over southern Germany, central Poland and the Ukraine to the Romanian Black Sea coast, temperatures high enough to
reach the top-10 are observed.

## 5 Drivers of the 1921 drought

### 5.1 1921 synoptic conditions

In this section, we explore the atmospheric circulation conditions associated with and contributing to the 1921 drought. Fig-
ure 12 shows the Z500 geopotential height monthly mean anomalies with the monthly precipitation as a ratio of the correspond-
ing 1981-2010 monthly climatology. Here we use the Z500 geopotential height as a 'proxy' of the sea-level pressure, where
high levels of the geopotential reflect high sea-level pressure values. This figure shows that a high-pressure system dominates
the 1921 circulation, although it moves around Europe from month-to-month. The high-pressure anomaly is situated roughly
over the British Isles and is present in Feb., April, June, July, Sept. and Oct. These months give anomalously low precipitation
values in western - and for some of these months - central Europe. In March, the high-pressure anomaly dominates all of
Europe, giving dry conditions in most of Europe except Scotland and western Norway in the north and southern Italy, Sardinia
and southern France/northwestern Spain in the south. North of the Black Sea, March lacks any rain. May is a typical saddle sit-
uation between troughs over Iceland and northwest Africa and ridges with their centres in eastern Europe and over the Atlantic.
This situation gives calm weather and does not ameliorate the precipitation deficit build-up in the preceding months. August,
with a weak low-pressure area north of Scotland brings some rain to parts of the UK and France, but from southeastern England
well into central and eastern Europe, the anomalously low precipitation amounts are sustained. The extensive high-pressure
centre north of Iceland in November gives dry conditions in much of Europe except the Balkan peninsula and areas north of it,



while December has a high-pressure centre east of Ireland extending into central Europe making that only the countries around the Baltic Sea and Denmark are wetter than usual.

Figure 13a shows the annual mean 1921 North Atlantic storm track density anomaly with respect to the ERA20C climatology, based on the tracking scheme developed by Hodges (1995) and using spherical kernel estimators (Hodges, 1996).

Figure 13 shows an evident poleward displacement of the Atlantic storm track which resulted in a large reduction of Atlantic storms (mid-latitude cyclones) into continental Europe and the UK. The overall reduction of storms is also striking over Southern Europe (ca. 60% decrease), extending into the central Mediterranean and to Eastern Europe.

Although smaller than its Atlantic counterpart, the Mediterranean storm track also contributes to the total number of storms and rainfall accumulated over Southern Europe. Mediterranean cyclones are generated on preferred cyclogenesis areas such as the lee of the Alps, the Gulf of Lyon and the Gulf of Genoa, where the land-sea contrast provides the right conditions for storm growth (Trigo et al., 1999). Mediterranean storms are mostly subsynoptic lows, triggered by the major North Atlantic synoptic systems (Trigo et al., 2002). Therefore, North Atlantic jet variability exerts a strong control also on Mediterranean cyclogenesis. It is plausible that the northward deviation observed for the 1921 Atlantic storm track contributed to the marked reduction of Mediterranean cyclogenic events during that year. Figure 13b shows the ERA20C genesis density climatology (contours) and the 1921 anomaly (shading), highlighting that the year of 1921 had about 60% fewer genesis events than a typical year.

The poleward displacement of the storm track is also well reflected in moisture transport anomalies. Figure 13c displays the anomalies observed for the annual mean Integrated Vapour Transport (IVT) (shading denotes anomaly in intensity, while arrows show the direction) and Z500 (contours). A notable north-south dipole of positive-negative IVT anomalies was observed between northwestern and southwestern Europe during 1921 (Fig. 13c). Although there were some seasonal differences, anticyclonic conditions persisted all year round, with increased frequency of ridges and blocking systems over the northeastern Atlantic and Western Europe, well reflected in the yearly weighted Z500 anomalies, and thus decreasing the zonal flow (and consequently moisture transport) towards Central Europe. Enhanced moisture transport was observed in northern sectors of the UK and in Scandinavia as a result of this anomalous synoptic pattern.

## 6   Discussion and Conclusion

There are several aspects that make the 1921 drought stand-out from other droughts. Perhaps the most intriguing is that this drought can be regarded as a 'compound event' (Zscheischler et al., 2018). The 1921 drought could have the impact it had because the preceding autumn 1920 and winter of 1920/1921 were dry as well; the conservative nature of drought carries the effects of the lack of precipitation in autumn and winter into the following spring and summer. Such chain of events, a preceding dry winter or a clustering of dry winters, have been linked to heightened summer drought risk for the UK (Marsh et al., 2007). While none of the seasons in 1920 and 1921 tops the scale of having the largest precipitation deficit on record, it is the combination of these seasons that makes the 1921 drought to such an extreme year.



The impacts of the 1921 drought are assessed by a systematic review of text-based reports from selected papers in five
countries where the reported drought impacts were classified into major impact categories, and by a more ad-hoc approach highlighting the major impacts in some other European countries. The focus on the impact is coupled to a reconstruction of daily temperature and precipitation of the 1920 and 1921 situation based on observations. This study shows that

1) the drought had its first major impacts in autumn 1920 in north Italy and started to affect other parts of Europe in spring 1921. The type of impact which was experienced first varies across Europe. There is more agreement on the type of impact
which lingered on the longest, that is the impact on water supply, and on agriculture and livestock farming.

2) The impacts on agriculture and livestock farming are reported on most frequently, except in the Czech Republic where wildfires (followed by problems with the water supply) were more dominant. This was also the case in Poland, where towns were completely or partially destroyed by fires.

3) The epicentre of the drought was observed in the area including Brussels, Paris and Lyon but a broad band from Ireland over
the UK, the Low Countries, the Greater Alpine Area to the east over Poland and the Ukraine was affected.

4) The reported watershortage is confirmed by the reconstruction of precipitation and (potential) evapotranspiration; over the catchement of the Seine river the number of rainy days was well below anything observed in the 1981-2010 baseline period while (potential) evapotranspiration values were up to twice as high as in the baseline period.

5) While the average summer or annual temperature in 1921 was not really remarkable for modern standards, the hottest days
in the summer of 1921 ranked in the top-5 of hottests days of the last 100 years for parts in west and central Europe. The soil desiccation will have been instrumental in generating the 1921 extreme temperatures in Europe, a mechanism which is shown to be a key player in more recent heat waves as well (Miralles et al., 2014; Sousa et al., 2020).

6) Instrumental to the 1921 drought was a persistent high pressure area situated roughly over the British Isles, although it moved around a bit from month-to-month. This synoptic situation steered away cyclones, shifting the Atlantic storm track
northwards.

The examined newspaper reports the UK and the Netherlands suggest a late spring frost in 1921 but this could not be confirmed in the meteorological measurements. A possible reason for this intriguing mismatch is that temperature measurements in 1921 in the Netherlands were made at 2.2m height whereas modern measurements are done at 1.5m. While this may bias this comparison, ground temperatures are hardly correlated to the air temperature at both these heights when stable conditions
in the atmospheric boundary layer occur. Our interpretation is that the newspaper reports probably relate to temperatures close to the ground which can be considerably lower than those at thermometer screen height. The persistent high pressure situation in spring 1921 favours calm and cloudless nights and the development of stable conditions in the boundary layer, leading to ground frost. In addition, crops may have developed faster because of the high afternoon temperatures of spring 1921 and may have been in a critical stage when the late frosts occurred, giving greater damage to the crop.

From the perspective of climate dynamics, there are similarities between the 1921 drought and other major droughts, especially in the atmospheric circulation patterns that give rise to the stable, sunny and dry weather found during spring and summer of 1921. Similar atmospheric conditions occurred in the 1976 and 2018 droughts. The north Atlantic Sea Surface Temperature



(SST) pattern of 1921 showed alternating cold and warm areas but a clear similarity with the patterns of the more recent major droughts seems absent.

The relation between cold anomalies in the North Atlantic surface waters and warm weather over Europe was the focus of Duchez et al. (2016), who highlighted the warm summer of 2015 in Europe. The exceptionally warm and dry summer of that year coincided with a persistent high-pressure system over Europe much in the same fashion as the 1921 drought. Duchez et al. (2016) argue that the driver for the high-pressure system over Europe is to be found in anomalously cool waters in the northern North Atlantic. It is not clear if similar arguments apply to the 1921 situation. However, the exceptional persistence

of the 1921 drought event suggest the existence of equally persistent drivers. This clearly is an area where more research is needed.

The drought of 1921 illustrates that a long historical perspective is valuable in assessing the full range of climatic variability. The remarkable nature of the 1921 drought - where a dry spring and summer was preceded by a dry autumn and winter - make this year a candidate for benchmarks in design or tests of water resource management strategies (Marsh et al., 2007).

The 1921 drought demonstrates the impact of a climatic extreme in a situation where the effects of climatic change on the hydrologic cycle and desiccation of the soils is limited. In the 2019 heat wave that hit Europe (Copernicus Climate Change Service, 2020), the soil moisture deficits were larger than what they would have been in the past decades, pointing to a climate change imprint Sousa et al. (2020). Would a similar synoptic situation occur again that gave rise to the 1921 drought, the impact of dry conditions on high temperatures will be stronger (Rasmijn et al., 2018) and societal impacts are likely to be higher as

well.

*Data availability.*  The E-OBS dataset documented here is available from the Copernicus Climate Change Service at https://surfobs.climate.copernicus.eu/dataaccess/access_eobs.php





**Figure 1.** Graphs showing reported impact type for the United Kingdom, the Netherlands, Belgium, Germany and the Czech Republic (top to bottom). The vertical axis shows the number of reports, the horizontal axis shows the date associated with the publication of the text-based report. Below each impact graph, the weekly sum of rainfall, averaged over the country and as the ratio with the corresponding climatological period, is shown where brown relates to lower than expected values. In the background, daily maximum temperatures, averaged over the country, are shown as deviations from the climatology. Temperature and precipitation data are based on the E-OBS dataset introduced in this study.





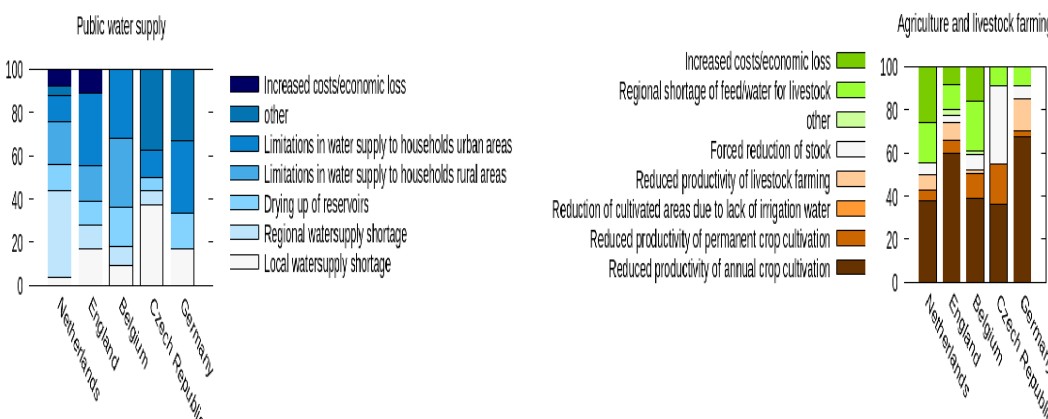

**Figure 2.** Reported impact types in Public water supply (left panel) and Agriculture and livestock farming (right panel) for the five countries.

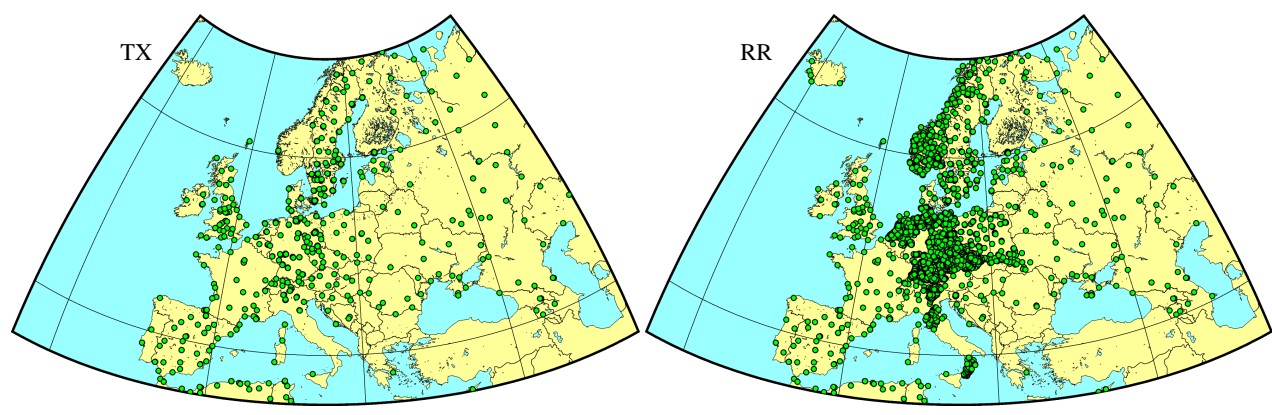

**Figure 3.** Map with the stations for daily temperature (left) and daily precipitation (right) that provide data for 1920-1921.





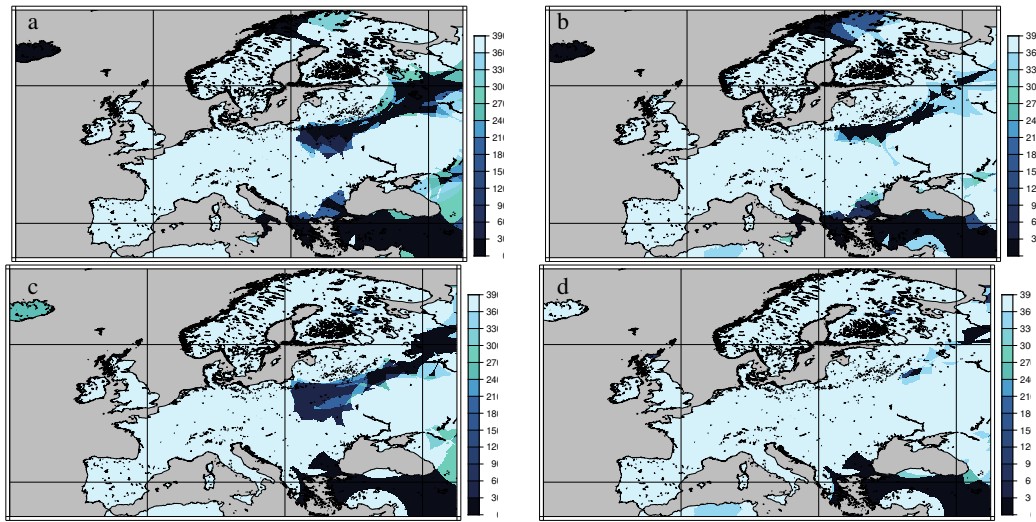

**Figure 4.** Maps indicating the number of days per year with available data for 1920 and 1921 (figs. a, c and b, d respectively) and for temperature and precipitation (figs. a, b and c, d respectively).

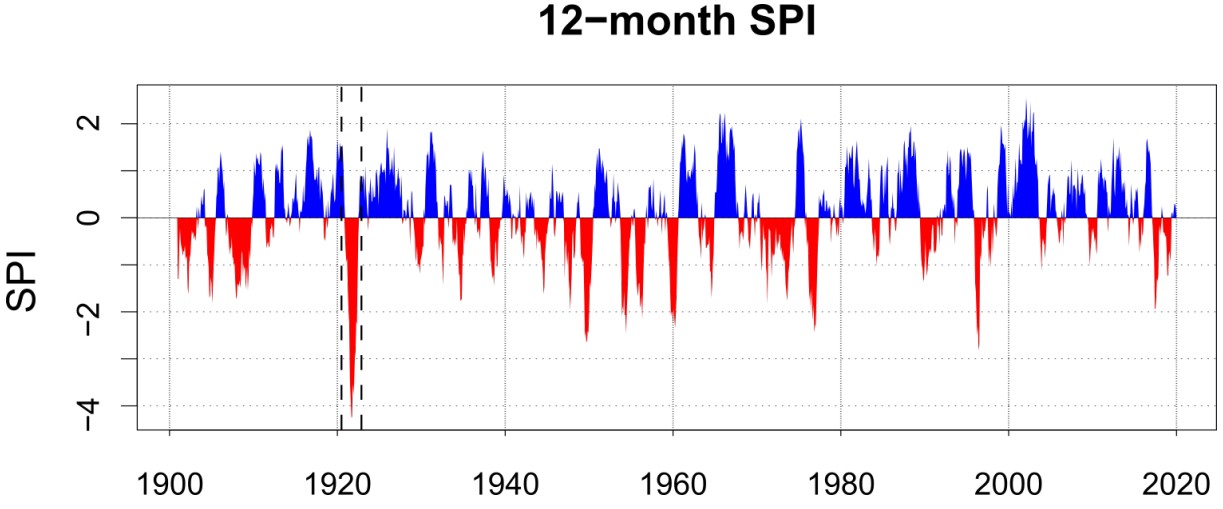

**Figure 5.** Monthly SPI12 values of the Uccle series for the period 1900-2019. Positive values (blue) show conditions wetter than usual, negative values (red) show conditions drier than usual with values below -2 relating to 'extremely dry'.



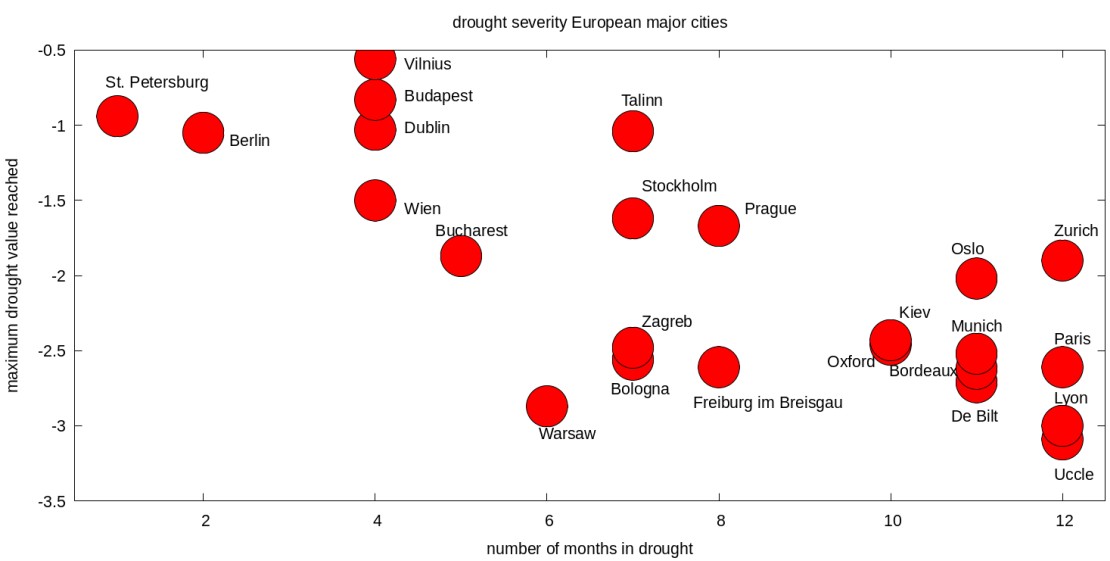

**Figure 6.** Diagram showing the drought severity of major European cities in 1921. The horizontal axis gives the number of months in 1921 that SPI3 values were below -0.5 and the vertical axis shows the lowest SPI3 value attained in 1921.

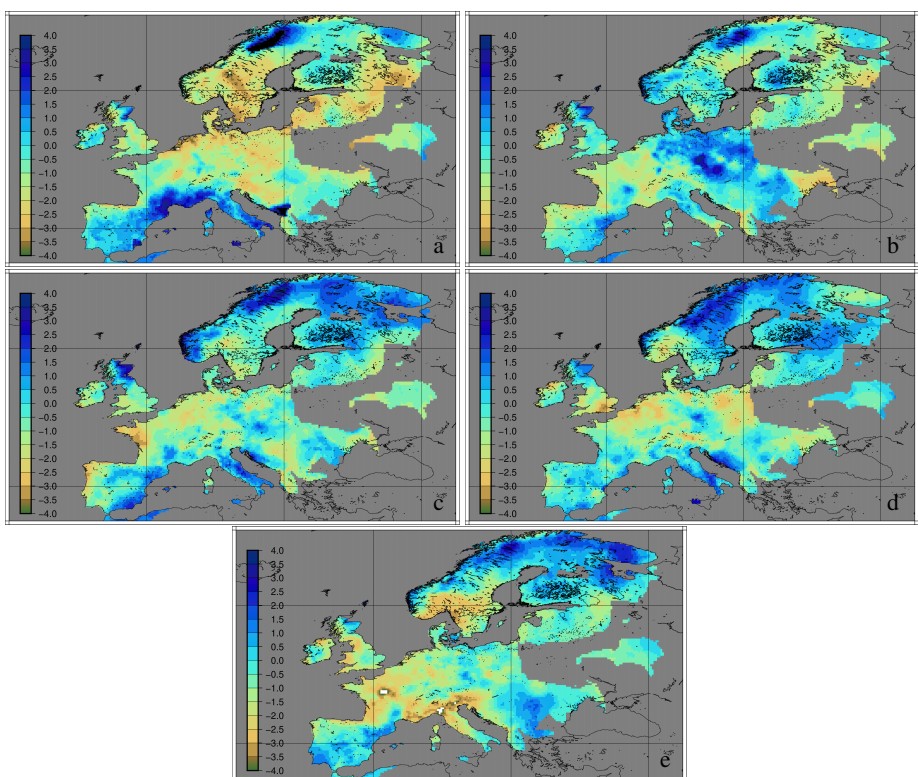

**Figure 7.** Maps with SPI values showing the evolution of the drought from autumn 1920 to autumn 1921. The panels show SPI3 values for Sep-Nov 1920 (a), Dec-Feb 1921 (b), Mar-May 1921 (c), Jun-Aug 1921 (d) and Sep-Nov 1921 (e). Maps based on the E-OBS data.

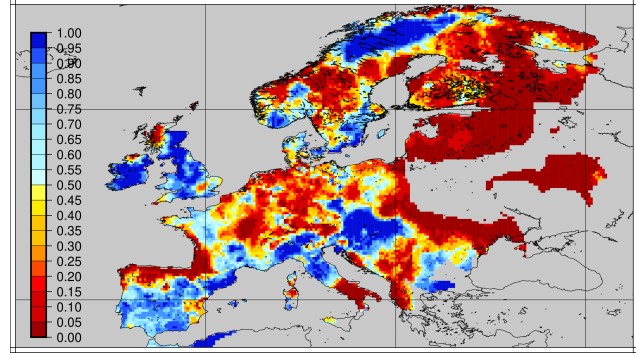

**Figure 8.** Probability-scaled map of the number of rainy days for 1921, using the 1981-2010 period as base period to calculate the expected value. The map shows the probability to observe the same number of rainy days in the 1981-2020 climate as was observed in the 1921 drought.




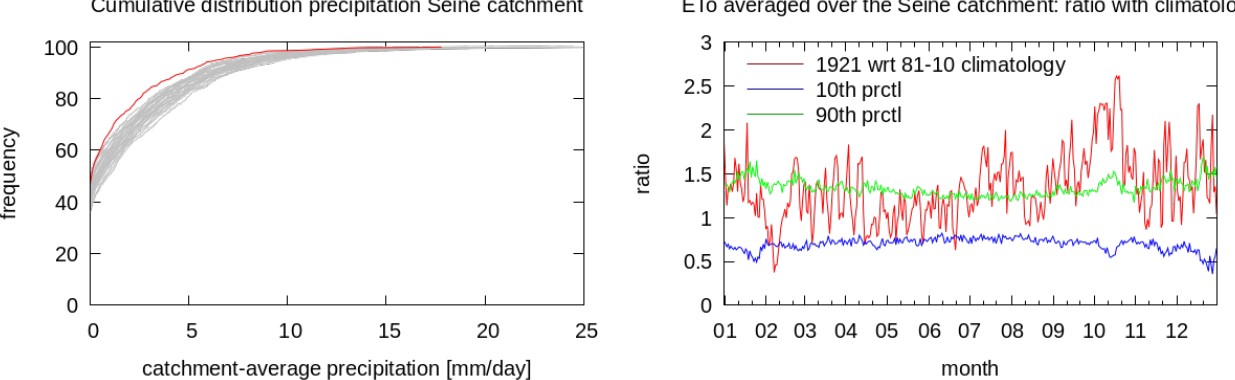

**Figure 9.** Cumulative distribution of daily rainfall aggregated over the Seine catchment (left panel). In red the data from the 1920-1921 period, the grey lines show individual years in the 1981-2010 period. The right panels show estimates of $ET_0$ for 1921 averaged over the Seine catchments as ratios to the 1981-2010 climatology. Added in blue and green are the 10[th] and 90[th] percentile of $ET_0$ values from the 1981-2010 climatology, again as a ratio from the mean value. Based on the E-OBS dataset.

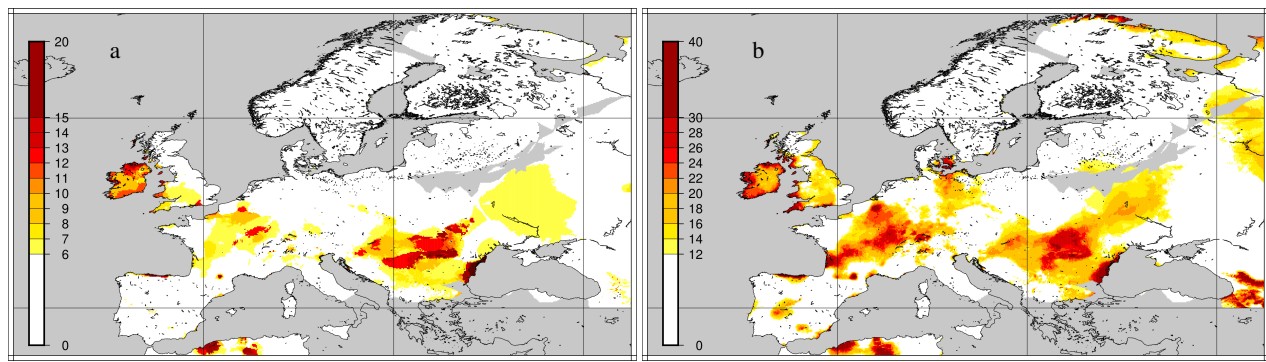

**Figure 10.** Maps showing (a); the Warm Spell Duration Index for summer 1921, a heat wave index relating to the locally defined threshold of the 90[th] percentile in daily maximum temperature and with a minimum length of 6 consecutive days above this threshold, (b) the number of days above this 90[th] percentile threshold, regardless if they are clustered in time or not. Based on the E-OBS dataset.



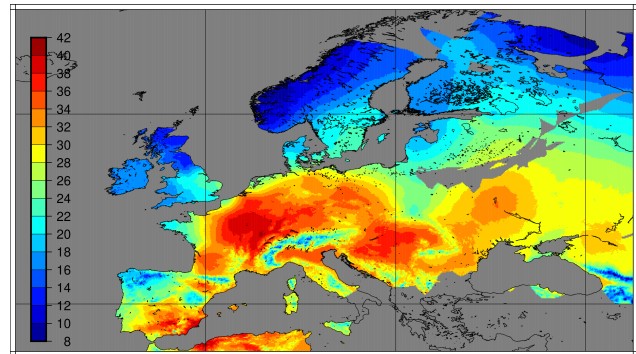

**Figure 11.** Map showing the daily maximum temperature for July 28, 1921 at the peak of the heat wave in France. Based on the E-OBS dataset.

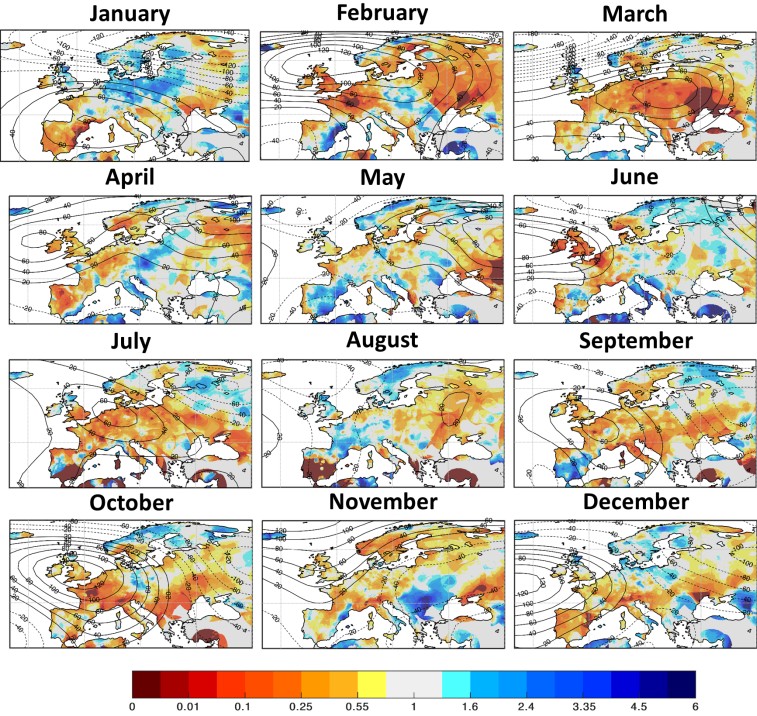

**Figure 12.** Monthly precipitation totals for January to December 1921 as the ratio of the corresponding 1981-2010 climatology. Contours show the Z500 anomalies (solid lines for positive, and dashed lines for negative, in 20m intervals).





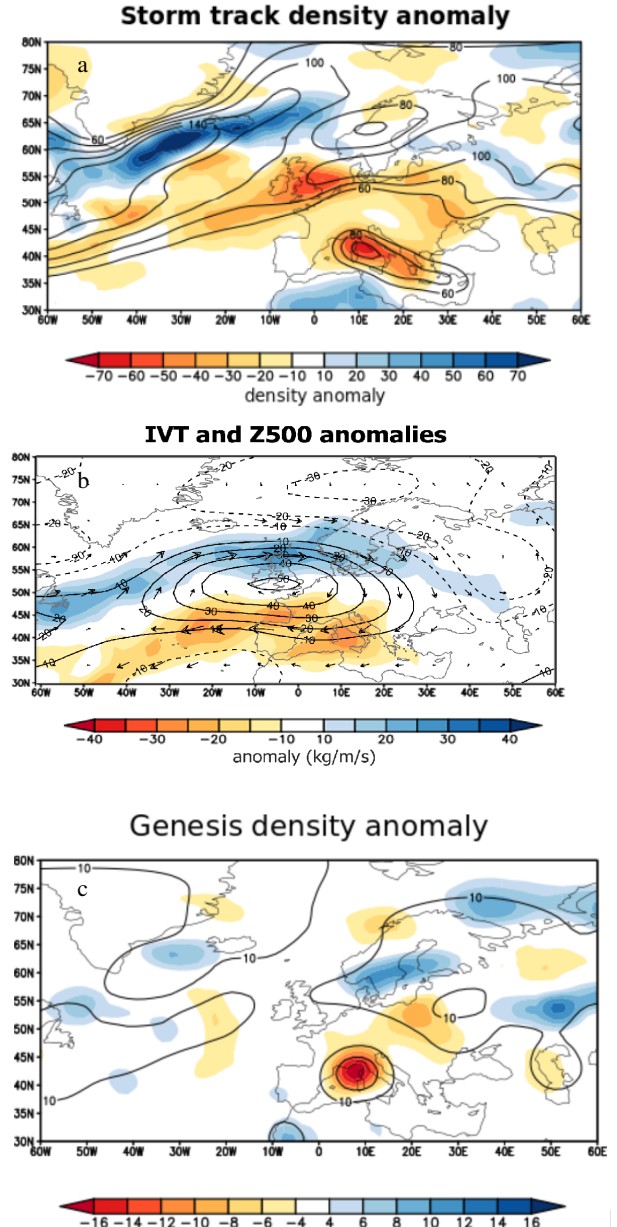

**Figure 13.** (a) 1921 storm track density anomaly (shaded) and storm track ERA20C density climatology (contours). Densities are in units of number of storm tracks per year, per unit area, where the unit area is equivalent to a 5° spherical cap ($\approx$ 106 km$^2$). (b) 1921 storm genesis density anomaly (shaded) and ERA20C genesis density climatology (contours). Densities are in units of number of storm tracks or genesis events per year respectively, per unit area, where the unit area is equivalent to a 5°spherical cap ($\approx$ 106 km$^2$ (c) 1921 vertically integrated horizontal vapour transport anomaly intensity( shading) and transport direction (arrows). Contours show the 1921 Z500 anomaly.




## Appendix A:  Photographical impressions of the 1921 drought

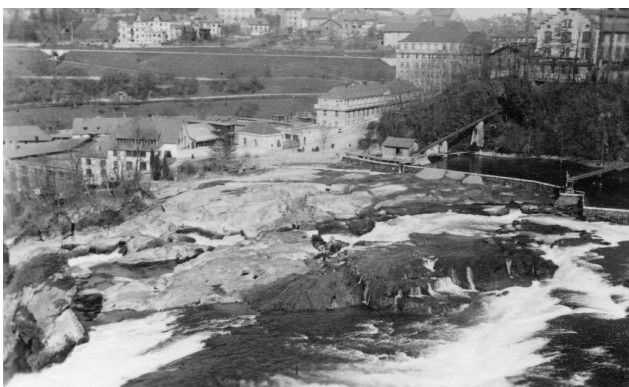

**Figure A1.** The Rhine Falls near Neuhausen (Switzerland) at low water in 1921. Photograph made on March 22, 1921 when the water level reached a historically low level. Source and copyright: Schweizerisches Socialarchiv.

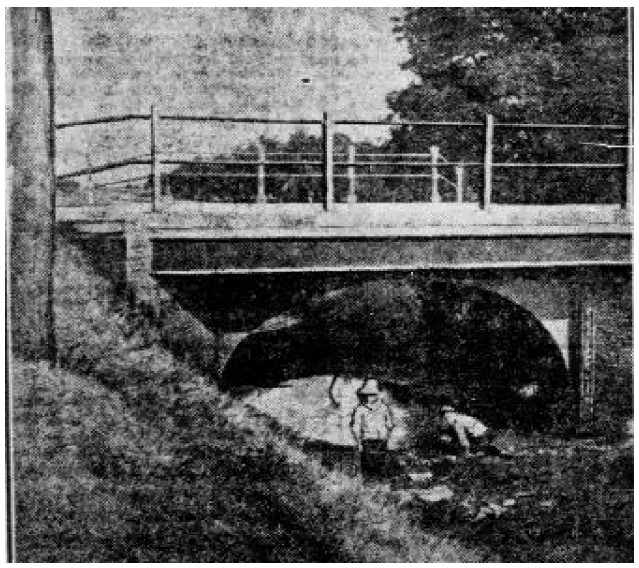

**Figure A2.** Photograph from the Dutch newspaper De Telegraaf, 1921/07/20. The original caption is: The drought. The river Linge holds since some time, as many of our rivers in our country, no water anymore. Only under the bridges is some water left and kids fill a jug to relief the water shortage of home.





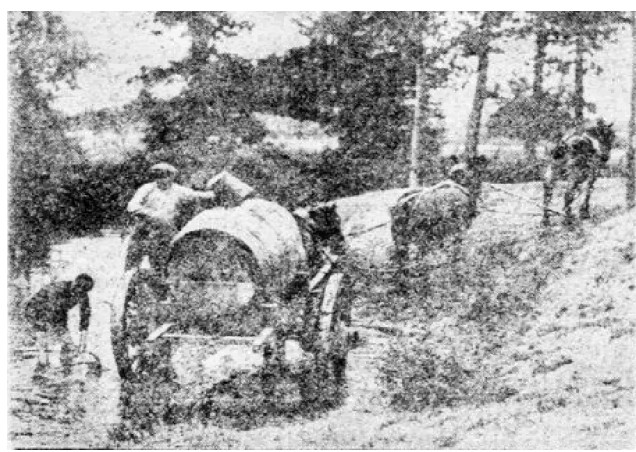

**Figure A3.** Photograph from the Dutch newspaper De Telegraaf, 1921/08/14. The original caption is: The drought in Zuid-Limburg. Fetching water from the Geul near Valkenburg. Water is sold against 25 cents a bucket or for a two-and-a-half guilder coin a cask and finds eager buyers.

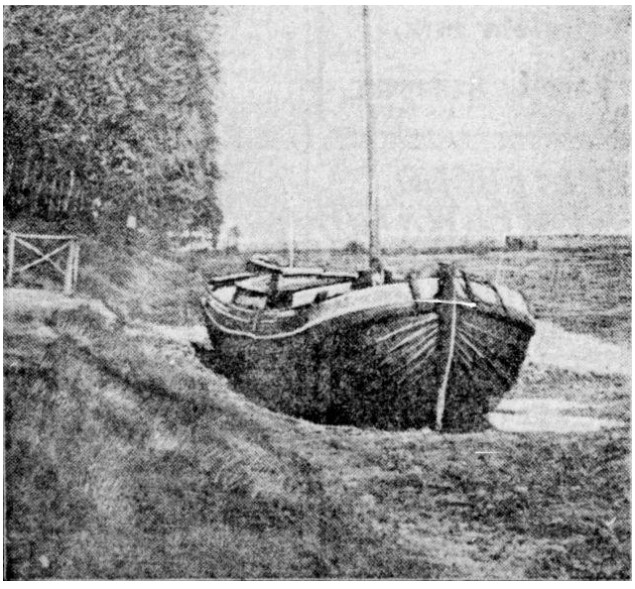

**Figure A4.** Photograph from the Dutch newspaper De Telegraaf, 1921/09/01. The original caption is: The drought in Friesland. In Friesland are several ships completely grounded because of the low waterlevel.



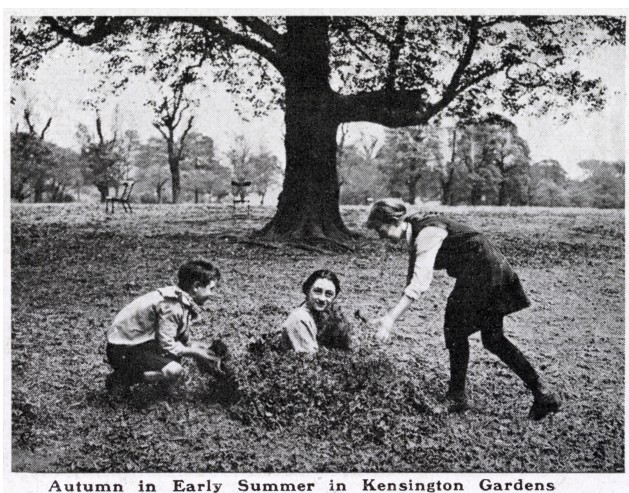

**Figure A5.** Photograph from the British illustrated newspaper The Sphere, 1921/07/30. The original caption is: Autumn in Early Summer in Kensington Gardens. Copyright: Illustrated London News Ltd/Mary Evans

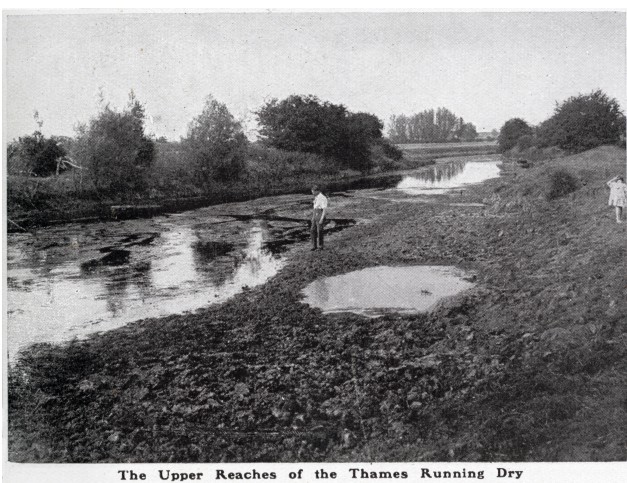

**Figure A6.** Photograph from the British illustrated newspaper The Sphere, 1921/08/13. The original caption is: The Upper Reaches of the Thames Running Dry. This picture was taken just above Lechiade in Glouchestershire. Usually at this time of year the river is very full of water. Now, however, it is nearly empty, and one can walk for long distances along what was formerly mid-stream. Copyright: Illustrated London News Ltd/Mary Evans





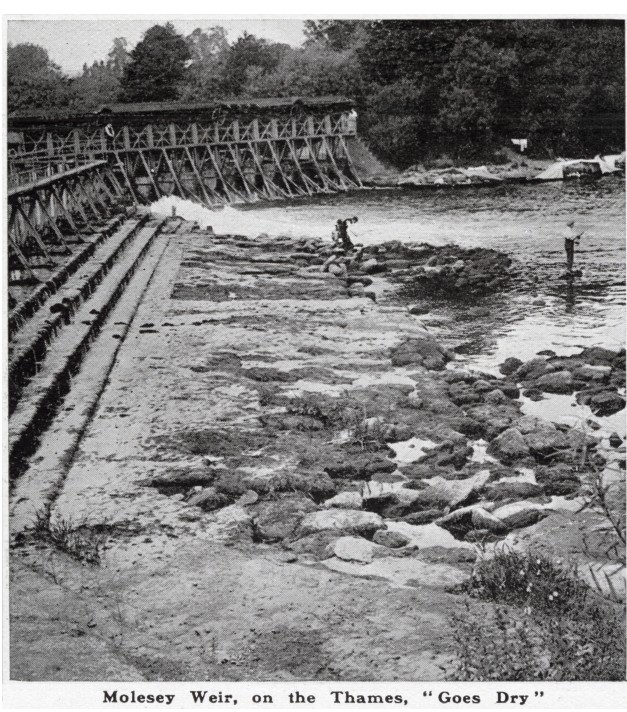

Molesey Weir, on the Thames, "Goes Dry"

**Figure A7.** Photograph from the British illustrated newspaper The Sphere, 1921/08/20. The original caption is: Molesey Weir, on the Thames, "Goes Dry". This picture gives a most vivid presentation of one of the effects of the drought. Molesey Weir, usually noted for its rush of water, is here seen almost perfectly free from water - a sight very rarely seen even by the "oldest inhabitants". Copyright: Illustrated London News Ltd/Mary Evans

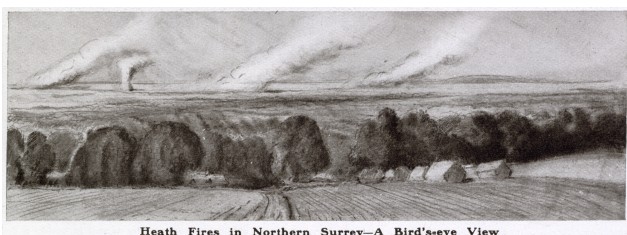

Heath Fires in Northern Surrey—A Bird's-eye View

**Figure A8.** Photograph from the British illustrated newspaper The Sphere, 1921/09/03. The original caption is: Heath Fires in Northern Surrey - A Bird's-eye View. This drawing was made on the North Downs. The panorama stretched away towards Aldershot on the west and St. George's Hill on the east. Frequently six or seven fires could be seen burning together at one time. Suddenly smoke would shoot upwards like a pillar, and then, meeting a current of air, it would drift away to the east. Copyright: Illustrated London News Ltd/Mary Evans





*Author contributions.* Meteorological station data used in the analysis and the production of the E-OBS were provided by R. Coscarelli,
A.A. Pasqua, O. Petrucci, M. Curley, M. Mietus, J. Filipiak, P. Štěpánek and P. Zahradníček. These authors also provided impact information
on the 1921 drought in their home countries. The E-OBS is produced by E.J.M. van den Besselaar. Surveys of newspaper articles were
performed by H. Van de Vyver, B. Van Schaeybroeck, R. Brázdil, L. Řezníčková and G. van der Schrier. The analysis of the atmospheric
circulation from the reanalysis was done by A. Ossó, P.M. Sousa and R.P. Allan. Analysis on the E-OBS data were performed by H. Van de
Vyver, B. Van Schaeybroeck and G. van der Schrier. The text was written by R.P. Allan, A. Ossó, P.M. Sousa, H. Van de Vyver, R. Brázdil,
G. van der Schrier, R. Trigo and E. Aguilar.

*Competing interests.* The authors declare that no competing interests are present

*Acknowledgements.* PŠ, PZ, RB and LŘ were supported by the Ministry of Education, Youth and Sports of the Czech Republic for the
SustES – Adaptation strategies for sustainable ecosystem services and food security under adverse environmental conditions, project ref.
CZ.02.1.01/0.0/0.0/16_019/0000797. The Project INDECIS is part of ERA4CS, an ERA-NET initiated by JPI Climate, and funded by
FORMAS (SE), DLR (DE), BMWFW (AT), IFD (DK), MINECO (ES), ANR (FR) with co-funding by the European Union (Grant 690462).

The production of the E-OBS is funded through the Copernicus Climate Change Service through contract C3S_311a_Lot4.

The newspapers articles from Algemeen Handelsblad can be access through www.delpher.nl, the Berliner Tageblatt is access through
zefys.staatsbibliothek-berlin.de. The Standaard is accessed through https://vlaamse-erfgoedbibliotheken.be/kranten/online, the Birmingham
Gazette through the www.britishnewspaperarchive.co.uk and the Lidové noviny through www.digitalniknihovna.cz/.



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
