# Peer review of "The 1921 European drought: Impacts, reconstruction and drivers"

_Climate of the Past, 2021_

## Author Response (AR1)

**reply to Reviewer 1, Dr. Olga Solomina**

Many thanks to the reviewer for going through the manuscript to assess its suitability for publication in Climate of the Past. The few remarks that are made on the language have been addressed.

Following the review, we added a reference to the European Russian Drought Atlas to emphasize (as we do in the Introduction) that the drought was extensive and reached up the Ural mountains.

In conclusion, we would like to stress that while the impact of the drought was strong in western and central Europe, the situation was more dramatic in eastern Europe. The data availability did not allow us to extent the reconstruction of daily precipitation and temperature further to the east. Hopefully there will be a new opportunity to reach out to other data holding archives to see if the reconstruction could be extended to the Ural mountains.

**Reply to reviewer 2**

Many thanks for reviewing the manuscript and bringing-up the concerns regarding the length and the comments on the wording and lack of explanation.

Specific comments (reviewer comments in italic):

*P1/L4: "(rescued) meteorological measurements" the term "rescued" is unclear here. Define or remove from the abstract*
It is removed from the abstract. The term is introduced in the manuscript further below.

*P1/L16: "The 1921 drought stands-out as the most severe and most wide-spread drought in Europe since the start of the 20th century. While none of the seasons in 1920 and 1921 tops the scale of having the largest precipitation deficit on record, the conservative nature of drought amplifies the lack of precipitation in autumn and winter into the following spring and summer" the meaning of this paragraph is not fully clear. The 21 drought stands out as the most severe on record, yet there are no deficits in observed precipitation??*
This was rephrased into: "The 1921 drought stands-out as the most severe and most wide-spread drought in Europe since the start of the 20th century. The precipitation deficit in all seasons was large, but in none of the seasons in 1920 and 1921 the precipitation deficit was the largest on record. The severity of the 1921 drought relates to the conservative nature of drought which amplifies the lack of precipitation in autumn and winter into the following spring and summer. "

*P3/L54: "systematic effort to review" provide more details on how this systematic literature search was conducted (e.g., keywords, publication dates, databases used, etc.).*
The information on the approach of the review was indeed incomplete and scattered in the manuscript. This paragraph has been rewritten into: "A systematic effort was made to review the impacts of the 1921 drought. This is done by collecting text-based reports from five digitized newspapers which reported drought impacts. The period for which the newspapers were reviewed was from January 1 to December 31 1921. Different selected newspapers were explored through their digital archives by means of the search term 'drought' (in each of the native languages). Text-based reports from the following digitized newspapers were used: the Birmingham Gazette (United Kingdom), the Algemeen Handelsblad (the Netherlands), the De Standaard (Belgium), the Berliner Tageszeitung (Germany) and the Lidové noviny (the Czech Republic). These papers can be accessed through their digital archives (webaddresses are provided in the acknowledgement).

The newspaper clippings were classified into major impact categories, each of which had a number of subtypes. The classification follows the pioneering

work of Stahl et al. (2016, their Table A2) and each report is labelled by area or place and date. The distribution of these categories and types was then analyzed over time for the five newspapers and taken to be representative for the related countries.

In addition to this systematic effort, a more ad hoc approach was taken for Ireland, northern Italy and the Alpine region, France and Poland. Newspaper-based reports on impacts and reported impacts in scientific publications from that period were used to identify impacts which were prominent in the reporting.

*P3/L66 please make sure that the figures in the supplement are placed in the same order as they are referred to in the main text.*
ordering of the figures is changed

*P3/L71: "BE and DE report" – define the abbreviations*
Abbreviations defined

*P7/L164: 0.4 mm – is this a common threshold? If so, provide a reference.*
The 0.4 mm was a common threshold used by the UK Met Office in that period. This remark has been added to the manuscript.

*P7/L172: "30°Cin" spacing needed*
corrected

*P8/L198 complement*
corrected

*P8/L204: "... amount of data that is used to construct the data" reword*
reworded

*P8/L216: I assume that the new met data were quality checked and homogenized before the gridding procedure? If so, this should be mentioned in the methods.*
The data are quality controlled. They are not homogenized. Although this was done in the context of the INDECIS project for the modern period, homogenization of daily data for the 1920s was not possible. This comment has been added to the manuscript.

*P9/L218 "[...] has been provided." provided from where?*
Source added.

*P9/L226 "daily temperature data" it should be mentioned here, or earlier in the methods, whether this refers to max, min or average daily temperature*
Comment added to make clear that with 'daily temperature data' we mean the set of daily tmax, tmin and daily averaged temperature data.

*P10/L261 RR1?*
Removed

*P10/L270: change to extraterrestrial or extra-terrestrial*
changed

*P11/L294: "For comparison, [. . . ]" if a comparison is made with fig 5, then please provided the values for SPI12 and not SPI3.*
Rephrased: a comparison between the SPI12 value of 1921 and more recent years is intended.

*P11/L297: "[. . . ] and peak values in 1967 are [. . . ]" unclear if these statistics are for SPI3 or SPI6 or SPI12?*
Rephrased

*P11/L301: "dry to extremely dry conditions" define the SPI threshold for these conditions*
Threshold values added to the text

*P14/L402: "[. . . ] largest precipitation deficit on record [. . . ]" unclear, which record this refers to.*
Rephrased and comments added to the text to clarify

*L408: "the drought had its first major impacts in autumn 1920 in north Italy" fig 7 seems to show rather pluvial conditions in northern Italy at that time . . .*
This was phrased awkward, the Alpine region saw the impacts in autumn 1920 and streams from the Alps to the north Italian plane were affected. Sentence is rephrased.

*Fig 4 caption does not make sense. What is the difference between panels a,b,c and d? Which of the panels represent temperature and which represent precipitation? Also, the last digits in the labels on the colorbars seems to be truncated? This figure is in my opinion better suited in the supplement .*
This figure is moved to the Appendix. One of the colour bars indeed had truncated numbers - changed. Labels have been added to the figure panels, making clear to which variable and what year each panels refers to.

*Fig 5: Indicate in the caption which country the Uccle series comes from (suggestion). Provide a title for the x-axis.*
In the caption it is made clear that Uccle is in Belgium. The x-axis of this figure clearly shows years - it would be superfluous to add a label.

*Fig 6 caption: x-axis – number of consecutive months?*
Yes - good point. Xlabel changed.

*Overall I find the labels for the colorbars in many of the figures too small to be clearly visible. Also, the figures would be easier to understand if a title would be added to the colorbars (e.g., in fig 4, 7, 8, 10, 11, 12)*

The mentioned figures are modified and now contain a title (except fig 4 - info added in the figure itself, and fig. 7 where titles would make the figure more messy). All mentioned figures now contain a title to the colourbar.